# The Role of Audiovisual Speech in Fast-Mapping and Novel Word Retention in Monolingual and Bilingual 24-Month-Olds

**DOI:** 10.3390/brainsci11010114

**Published:** 2021-01-16

**Authors:** Drew Weatherhead, Maria M. Arredondo, Loreto Nácar Garcia, Janet F. Werker

**Affiliations:** 1Department of Psychology and Neuroscience, Dalhousie University, Halifax, NS B3H 4R2, Canada; 2Department of Human Development and Family Sciences, University of Texas at Austin, Austin, TX 78705, USA; maria.arredondo@austin.utexas.edu; 3Department of Psychology, University of British Columbia, Vancouver, BC V6T 1Z4, Canada; loretonacar@edai.cat (L.N.G.); jwerker@psych.ubc.ca (J.F.W.)

**Keywords:** word learning, fast-mapping, audiovisual speech perception, word recognition

## Abstract

Three experiments examined the role of audiovisual speech on 24-month-old monolingual and bilinguals’ performance in a fast-mapping task. In all three experiments, toddlers were exposed to familiar trials which tested their knowledge of known word–referent pairs, disambiguation trials in which novel word–referent pairs were indirectly learned, and retention trials which probed their recognition of the newly-learned word–referent pairs. In Experiment 1 (*n* = 48), lip movements were present during familiar and disambiguation trials, but not retention trials. In Experiment 2 (*n* = 48), lip movements were present during all three trial types. In Experiment 3 (bilinguals only, *n* = 24), a still face with no lip movements was present in all three trial types. While toddlers succeeded in the familiar and disambiguation trials of every experiment, success in the retention trials was only found in Experiment 2. This work suggests that the extra-linguistic support provided by lip movements improved the learning and recognition of the novel words.

## 1. Introduction

One of the hallmarks of language acquisition is the remarkable ease with which children acquire new words. From a very young age, infants are sensitive to both the acoustic and visual properties of spoken language [1,2]. By 6 months, infants recognize highly familiar words such as their name [3,4], and by 17–18 months, they map unfamiliar words to novel items [5]. To become masters of their native language(s), it is imperative that children determine the mappings between words and their referents, yet evidence suggests that children may not often retain word–referent mappings. Furthermore, bilingual-learning toddlers may not use similar word–referent mappings as a word learning strategy in a similar manner to monolinguals, since they regularly learn two words for one referent. The present studies investigate whether audiovisual cues support the retention of a new word in both bilingual and monolingual toddlers.

### 1.1. The Road to Fast-Mapping May Not Lead to Retention of a New Word

Under a lexical constraints account, children assume that mappings from wordforms to referents are mutually exclusive and, as such, have only one referent [6,7,8]. Thus, when young toddlers (~17 months) hear a new word, they often reject objects with known labels (e.g., ball, car) as possible referents and infer that the new word belongs to a novel or unfamiliar object if one is available [5,9]. This word-learning strategy is known as disambiguation, and the word–referent mappings acquired through disambiguation are known as fast-mapping. Although young toddlers can map novel words to referents, they do not show retention of these novel word–referent mappings until 36 months [10]. However, evidence shows that children’s vocabulary rapidly grows between 12 and 24 months of age [11,12], thus casting doubt that children are unable to retain new words as quickly as they are encountered.

A dynamic associative approach, however, suggests that fast-mapping is emergent from a two-system word learning process that entails rapid referent selection and word recognition, as well as slow associative learning that supports the retention of newly learned words [13,14]. In that regard, several studies show that 24-month-olds are able to retain new word–referent mappings when there are ostensive signals (e.g., reinforcement), or when there is silent familiarization to the target referent before the fast-mapping task [14,15,16]. In the absence of exposure and ostensive support, many studies show that 24-month-olds are unsuccessful at retaining these word–referent mappings [10,12] and that not all types of ostensive support lead to retention at this age (e.g., [15]). For example, in Horst and Samuelson (2008), 24-month-olds retained fast-mapped words, during disambiguation trials, only when their referent selection was reinforced (e.g., “Yes that is the cheem”) [14]. The authors suggest that without the ostensive support, the word–referent mappings formed through disambiguation are too weak to support encoding. One exception to these findings is Kalashnikova et al. (2018), in which monolingual and bilingual toddlers disambiguated and retained novel object–label pairs at 18 months of age, but only monolinguals showed retention at 24 months [17].

Why might bilinguals fail at retaining labels for a new word–referent pair? A growing body of evidence suggests that bilingual toddlers under the age of 2 are less likely to show disambiguation, as compared to their monolingual peers [18,19]. Bilingual children’s weaker fast-mapping appears to be related to experience with their two languages [20]. For instance, 18-month-old bilingual toddlers who know a greater number of translational equivalents (i.e., a label in each of their languages for the same referent) are less likely to show disambiguation [18] (see also [17]). Bilingual children also use disambiguation less robustly with age (e.g., 3–4 vs. 5–6 years old; [21,22]). In contrast, monolinguals’ use of disambiguation increases with age, and their greater use of this heuristic is associated with greater vocabulary knowledge [5,10]. Currently, the extent to which bilinguals use fast-mapping as a viable word learning strategy, and how dual-language experience and vocabulary within and across their languages plays a role, is still highly debated [17,18,19,20,21,22].

The “lexical constraints” and the “dynamic associative” accounts differ in their explanations for how children’s linguistic experiences (i.e., how word–referent mappings operate in their environment) impact whether a novel word is mapped to a novel referent and whether these words are retained. Under both of these accounts, there are two possibilities that explain the discrepancies between monolinguals and bilinguals. A lexical constraints account would suggest that while both monolingual and bilingual 24-month-olds disambiguate, the degree to which they rely on disambiguation may differ as a function of their previous experience. That is, mutual exclusivity may be a viable word learning strategy for monolinguals given their common experience with a one-to-one mapping between labels and object categories; however, it may not be a highly reliable word learning strategy for bilinguals, given their common experience with at least two labels for each category, one in each language [17,21,23,24]. Under the lexical constraints account, then, if bilinguals rely less on fast-mapping as a viable word learning strategy, disambiguation of novel words, and retention of those disambiguated meanings, would not occur. Alternatively, a dynamic associative account would suggest that bilingual toddlers’ increasing vocabulary in both of their languages simultaneously also increases difficulty in the complexity of the task. That is, while the phenomenon that supports rapid referent selection may yield the same results for monolinguals and bilinguals, the phenomenon underlying the slow associative learning leading to retention may require additional time and support for bilinguals. Under this interpretation, retention would still be observed for bilingual 24-month-olds if additional linguistic support is provided. Nevertheless, there are still a number of studies that suggest that even monolingual 24-month-olds do not retain fast-mapped word–object pairs [10,14]. As such, additional linguistic support could enhance retention in monolingual toddlers as well. Audiovisual cues are one example of linguistic support that children rely on during language acquisition; we turn to this important scaffolding next.

### 1.2. Using Visual Speech Information as Additional Support for Indirect Word Learning

Viewing the lip movements from a speaker provides critical information about temporal and phonetic properties of the acoustic signal during speech perception (e.g., [25]). Even very young infants are sensitive to how speech maps onto visual articulatory gestures [2,26,27,28,29]. Lip movements, in turn, can augment speech processing and comprehension in both adults [30,31] and in infants [32]. For instance, infants and young children look at a speaker’s mouth when disambiguating speech forms that are difficult to distinguish in speech processing [25]. This is especially evident in work demonstrating that bilinguals learning languages that are rhythmically and phonologically similar (such as Spanish and Catalan) are more likely to attend to a talker’s mouth than bilinguals learning languages that are more rhythmically and phonologically distant (such as Spanish and Russian) [33]. Young children also attend to the speaker’s mouth as a form of scaffolding to distinguish wordforms that are acoustically similar (see [34]). For instance, when 12-month-old infants are exposed to mispronunciations of familiar words, such as an altered phoneme, that are visually consistent or inconsistent with the acoustic phoneme (i.e., the lip movements for “pottle” look like “bottle”, whereas the lip movements for “dottle” do not), they recognize the visually consistent mispronunciations (“pottle”), but not the visually inconsistent mispronunciations (“dottle”), even when the audio in both cases is unambiguously mispronounced. These findings suggest that when infants view mouth movements consistent with a known word, this can facilitate their word recognition.

Specific to word learning, 18-month-old infants have been shown to successfully learn word–referent pairs when they are taught the label verbally (i.e., only hear the label) and later recognize the auditory-learned labels in both the auditory modality (i.e., only heard the word) and visual modality (i.e., only saw a face articulating the word). In this same study, however, 18-month-olds failed to learn word–referent pairs when labels were taught in the visual modality (i.e., only saw the face articulating) and did not recognize the visually-learned labels in either modality [35]. While adults succeed in the visual-only learning condition [35], it is not until 30 months old that toddlers are able to learn new word–referent pairs when the label is presented only in the visual modality [36], and even then, toddlers struggled to recognize the visually-learned labels later in the auditory modality; that is, they recognized the labels based solely based on lip movements and did not recognize the labels when they were said. In both studies, the words were taught in an unambiguous word learning situation in which only one object was present on the screen during labeling.

Across development, children also vary in the amount of time they spend attending to facial cues, and this may also depend on the child’s experience with language. Prior to 6 months, infants tend to attend to the eyes and then gradually start to attend to the mouth when observing dynamic faces [37]. When listening to their native language, infants continue to attend to the mouth until approximately 10–12 months old, when attention begins to shift primarily to the eyes. When listening to a foreign language, however, 10–12-month-olds continue to focus on the mouth [25] (consistent with adult findings [38]). As infants’ experience with their native language increases, the need to focus on visual cues of spoken speech decreases [25,39,40]. The notion behind the shift is that attention to the mouth is driven by an attempt to learn language, a notion that is also supported by work showing that bilingual infants attend more to the mouth than monolingual infants throughout development [41] and show the same bias when scanning dynamic faces that show different emotions but not linguistic information [42]. These studies support the hypothesis that as bilingual infants face the challenge of simultaneously learning two rhythmically and phonetically close languages, they may rely more heavily on lip movements to assist them during discrimination and learning of their two languages, and for a longer period of time during development [33]. However, recent studies have found no differences in developmental patterns between monolinguals and bilinguals learning more distant languages, such as French and English [43]. In this case, peak attention to the mouth was found at 24 months for both groups of toddlers, which lasted until their fifth birthday. However, in these studies, attention to the mouth was recorded while infants and children watched videos of speakers saying a monologue; there was no word recognition or word learning task occurring. Whether audiovisual speech affects word learning in monolingual and bilingual toddlers remains to be seen.

In sum, infants and toddlers are highly sensitive to visual speech information, and importantly, the presence of visual speech information can influence word processing in infant and adult populations [32,44,45].

### 1.3. The Present Study

In the present work, we ask whether the presence of visual speech information supports toddlers’ retention of novel words during a fast-mapping task. We assessed monolingual and bilingual toddlers’ ability to both disambiguate and retain referents for novel labels at 24 months. Monolinguals had been exposed to a primary language for at least 90% of their daily life, and bilinguals had been exposed to a primary and secondary language for at least 30% in their daily life. In Experiment 1, 24-month-old toddlers completed a fast-mapping task in which they had access to both auditory and visual speech information during labeling. When testing for word retention, infants only heard the labels and did not receive visual information. In Experiment 2, toddlers had access to both auditory and visual speech information during both labeling and retention, to determine whether the recognition of the object–label pairs was modality-dependent. To more specifically determine whether it was indeed the visual speech information that enhanced retention versus the simple presence of a face, in Experiment 3, we showed a static image of face during labeling and retention.

We hypothesized that audiovisual information would support toddlers’ word retention, regardless of language background (monolinguals and bilinguals). Given prior evidence suggesting that infants’ attention to the mouth is associated with greater vocabulary knowledge, we also explored whether attention to the mouth during disambiguation trials would be associated with stronger retention scores. Additionally, in all experiments, we recorded children’s productive vocabularies, since previous work has demonstrated conflicting evidence of whether productive vocabulary is correlated with better retention of new words in a fast-mapping task. In Bion et al. (2013), productive English vocabulary was strongly correlated to monolinguals’ looking to the novel object during disambiguation trials [10]. However, in Kalishnakova et al. (2018), vocabulary was not correlated with disambiguation scores for either monolingual or bilingual toddlers [17]. Finally, as previous work has suggested that infants and toddler’s attention to the mouth is driven by an attempt to learn language [37,38,39,40,41], we recorded children’s looking to the speaker’s mouth when present for the purpose of exploratory analyses, the rationale being that greater attention to the mouth will be paid when an unfamiliar object(s) is on screen, as there is an opportunity for learning (disambiguation trials, Experiment 1 and 2), and additional information from the mouth may be needed for accurate recognition (retention trials, Experiment 2). We assessed whether there were correlations between looking to the mouth when there were 0, 1, and 2 unfamiliar objects on the screen and children’s retention of the fast-mapped words.

## 2. Experiment 1

### 2.1. Participants

Forty-eight 24-month-old toddlers (24 females, 24 males) took part in the experiment. Of these, 24 were being raised monolingual (M_age_ = 24 months 2 days, range (years; months; days old) = 1; 11; 10–2; 0; 23) and 24 bilingual (M_age_ = 24 months 4 days, range: 1; 11; 11–2; 0; 17; see Appendix A for information about language background) from birth. See Table 1 and Table 2 for vocabulary scores and average language exposure across experiments. An additional 10 children were tested but excluded from analyses due to fussiness (2), failure to attend to the screen (2), equipment failure (3), experimenter error (1), and for not meeting the language requirements of the study (2). All participants were born full term (≥37 weeks) and had no known hearing or vision problems. Families were recruited from the Early Development Research Group database at the University of British Columbia, which recruits new parents at a maternity ward (in Vancouver) at the time of the child’s birth.

### 2.2. Procedure

Children were tested in a dimly lit, sound-attenuated room at an infant laboratory. Children sat on their parent’s lap and faced a computer monitor at an approximate distance of 60 cm. The experimenter controlled the study through a stimulus presentation computer in a screened-off area of the same room. Parents were instructed to refrain from interacting with their child during the experiment, wore opaque glasses to reduce bias, and listened to music through headphones to mask the audio being played to the child.

Child eye-gaze data were collected using an eye tracker. The study started with a 5-point calibration routine to establish each child’s eye characteristics and location. Following successful calibration, each toddler was presented with the experiment. After the testing session, parents completed questionnaires regarding the child’s language abilities, as well as the family’s language and demographic backgrounds. At the end of the visit, children received an honorary degree certificate, a lab newsletter, and a child size t-shirt as a thank you for their participation.

### 2.3. Measures

#### 2.3.1. Demographics

Parents reported on their child’s health, birth weight, sex, and gestational age. Parents also reported their level of education, work status, and racial or ethnic background. In all three experiments, parental education levels were on average between a Bachelor’s degree and a Master’s degree (see Table 1 for a full summary).

#### 2.3.2. Vocabulary Report

Parents completed the MacArthur-Bates Communicative Development Inventory (MCDI) Words and Sentences short form in English [46]. Parents whose child had exposure to more than one language were asked to mark items in each language using the same English form (see Table 2). Bilingual children in all three experiments received no systematic exposure to a third language.

#### 2.3.3. Language Exposure

Parents completed a one-on-one questionnaire with the experimenter (modified from [47]; see also MAPLE [48]), in which they reported on their child’s language exposure by speakers who spent significant time with them (e.g., at home, daycare, library, playgroups). This assessment allowed for a percentage calculation of how often each child was exposed to English versus another language (see Table 3).

#### 2.3.4. Indirect Word Learning (Fast-Mapping) Task

Children completed a word learning task that followed a similar procedure to Bion et al. (2013) [10]. The task included three types of trials: familiar, disambiguation, and retention. A total of 28 trials were presented: the disambiguation trials (12 total) appeared before the retention trials (8 total), and the familiar trials (8 total) were interspersed with disambiguation trials and retention trials throughout the task in a gradient manner.

The experimenter controlled the study through the stimulus presentation computer out of sight from the infants. Each trial started with a looming and colorful circle in the middle of the screen that changed colors and acted as an “attention getter”. Testing trials began immediately once the child’s attention was captured and focused to the center of the screen. On each trial, two objects first appeared in silence for 2.5 s (pre-naming), one object on the center left side of the screen and another object on the center right side of the screen. During familiar and disambiguation trials, a video of the face of a White female facing front appeared at the top center of the screen. Next, the video of the actor producing the naming phrase played (for 2–3 s) and disappeared once the phrase was completed. During retention trials, only the sound file of the naming phrase (no video) played. Finally, the objects remained on the screen for an additional 2.5 s of silence (post-naming). The objects were displayed during the entire trial, which lasted a total of 8 s. The total duration of the task was approximately 4 min.

##### Visual Stimuli

The visual stimuli were pictures of six familiar objects that represented some of the first words learned by infants exposed to English (car, cup, book, ball, cookie, shoe) and two novel objects (see Bion et al., 2011). The two novel objects were images downloaded from the TarrLab Object DataBank [49], with the criteria that they did not resemble any familiar objects in our stimuli or any other likely known objects at this age. All objects were controlled for size and perceptual similarity. Arrays of two objects were presented on a grey background, these objects were automatically located by the SMI begaze software in the middle-left or middle-right positions of the screen, at the same distance from the center of the screen (See Figure 1). Arrays consisted of two familiar objects (familiar trials only), one familiar object and one novel object (familiar trials and disambiguation trials), and the two novel objects from the disambiguation trials (retention trials only). Arrays were counterbalanced such that each object, and each object type (familiar vs. novel), appeared on each side of the screen in equal proportions.

##### Audiovisual/Audio Stimuli

Video and auditory-only recordings of the naming phrases were recorded in infant-directed speech by a female native speaker of Canadian English producing the eight target words, two novel (dofa, modi) and six familiar (cup, ball, shoe, car, book, cookie) words in three carrier phrases (“where is the [target word]?”, “look at the [target word]” and “find the [target word]”). Children saw all six familiar objects during familiar and disambiguation trials, although not all were named (contingent on their counterbalancing condition). Stimuli were recorded in a sound-treated room and were later equated for amplitude in Praat [50].

##### Apparatus

Eye gaze data were collected using a SMI Red250Mobile eye-tracking system that sampled at a rate of 60 Hz. The task and eye-tracker were controlled from a Dell Precision laptop and programmed and presented using SMI Experiment Suite 360 software. The task was presented to the participants on a 22″ LCD color display monitor. Auditory stimuli were played at approximately 65 dB via a computer speaker located behind the monitor screen. A Sony Handycam camcorder was placed on top of the monitor screen to record the child’s behavior in real time and monitor behavior to the screen.

#### 2.3.5. Analysis

##### Gaze Coding

Children’s fixations to the objects (center left, center right) were recorded during the task. In addition, during the familiar and disambiguation trials, fixations toward the speaker’s mouth were also recorded. Finally, the total duration of fixation toward the target object was calculated for pre-naming, naming, and post-naming periods. All eye-tracking data were processed using the SMI BeGaze software. The raw average rate of gaze loss during trials was 14.3% across all experiments (14.6%, 14.1%, and 14.3% for Experiments 1, 2, and 3, respectively). Trials in which there were no eye-tracking data during either the pre-naming or post-naming phase were excluded from our analysis. See below for more information.

##### Mouth

A proportion score was calculated for each familiar and disambiguation trial to determine the amount of time children spent attending to the mouth of the speaker during the naming period, in relation to the two objects on the screen.

##### Trials

Looking proportions to the target object for each trial were determined for the pre-naming period and for the post-naming period. Specifically, the post-naming period began 300 msec after the onset of the target word to account for the time infants and children take to shift their eyes in response to the word (e.g., [51]). The post-naming window spanned 2.5 s. Post-naming windows of 2.5 s and longer are typical in word learning studies using looking time measures with toddlers [10,52,53]. A difference score was calculated for each trial using the looking proportions for each period (proportion target object_postnaming_-proportion target object_prenaming_) [17]. This measure indicates the change in looking towards the target object after labeling. A positive score indicates increased looking to the target, while a negative score indicates looking to the other object.

Each trial type (familiar, disambiguation, and retention) was analyzed separately, as the difference score for each trial type was based on a different criterion (i.e., differing target objects).

##### Exploratory Analysis

For the purposes of investigating exploratory correlations between looking to the mouth and looking to the target object, we distinguish between two types of trials in which attention was paid to the mouth: trials in which there were no novel objects on the screen (i.e., familiar trials in which there were two known objects such as book and cup), called No Learning trials (NL), and trials where there was one novel object on the screen (familiar and disambiguation trials in which there was one familiar object and one novel object on the screen), called Potential Word Learning trials (PWL). In total, there were 4 No Learning Trials and 16 Potential Word Learning Trials. Attention to the mouth during No Learning and Potential Word Learning Trials was compared to determine if there were any overall differences across trial types. Furthermore, correlations between looking to the mouth during PWL trials and performance on disambiguation and retention trials were run.

Additionally, for disambiguation trials, we assessed whether toddlers’ vocabulary scores were correlated with disambiguation scores (consistent with [10]). For the bilingual toddlers, we also assessed whether the number of translational equivalents was correlated with disambiguation scores (consistent with [19,20]).

### 2.4. Results

#### 2.4.1. Familiar Trials

One-sample *t*-tests indicated that both monolingual and bilingual toddlers’ looking increased to the target object following the labeling event (*t*(23) = 3.23, *p* = 0.004, *d* = 0.659, and *t*(23) = 3.62, *p* = 0.001, *d* = 0.740, respectively; Figure 2). Independent *t*-tests comparing monolinguals and bilinguals revealed that monolinguals’ looking increased to the target object more than bilinguals, *t*(46) = 0.96, *p* = 0.028, *d* = 0.275.

#### 2.4.2. Disambiguation Trials

One-sample *t*-tests indicated that both monolingual and bilingual toddlers’ looking increased to the target object following the labeling event, *t*(23) = 2.18, *p* = 0.040, *d* = 0.445, and *t*(23) = 2.41, *p* = 0.025, *d* = 0.491, respectively. Independent *t*-tests comparing monolinguals and bilinguals revealed no difference in increased looking to the target object *t*(46) = 0.135, *p* = 0.954.

For exploratory analyses, we assessed the relation between English vocabulary and performance during disambiguation trials (see Table 2 for descriptive information regarding vocabulary scores). Using Pearson correlations, we found that English vocabulary significantly correlated with monolinguals’ performance in the disambiguation trials, *r*(24) = 0.50, *p* = 0.014, but not for bilinguals, *r*(24) = −0.13, *p* = 0.542. This lack of correlation held true for the bilinguals even when total vocabulary across both languages was considered *r*(24) = −0.15, *p* = 0.485. For the bilingual toddlers, we also assessed whether the number of translation equivalents was correlated with disambiguation scores, but there was no significant correlation, *r*(24) = −0.12, *p* = 0.564.

We also measured whether looking to the mouth during Potential Word Learning trials (i.e., those where there was one novel object and one familiar object) was correlated with disambiguation scores; however, this was not true for monolinguals, *r*(24) = −0.34, *p* = 0.103, or bilinguals, *r*(24) = −0.11, *p* = 0.609.

#### 2.4.3. Retention Trials

One-sample *t*-tests indicated that neither monolingual or bilingual toddlers’ looking increased to the target object following the labeling event, *t*(23) = 0.89, *p* = 0.384 and *t*(23) = 1.12, *p* = 0.273, respectively. Independent *t*-tests comparing monolinguals and bilinguals revealed no difference in looking to the target object, *t*(46) = 0.37, *p* = 0.510.

Additionally, we measured whether looking to the mouth during Potential Word Learning trials was correlated with retention scores; however, this was not true for monolinguals, *r*(24) = −0.05, *p* = 0.800, or bilinguals, *r*(24) = 0.16, *p* = 0.443.

#### 2.4.4. Attention to Mouth

A mixed-measures ANOVA with within-factor Trial Type (mouth NL vs. mouth PWL) and between-factor Language (monolingual vs. bilingual) revealed no significant main effects, Trial *F*(1, 46) = 1.64, *p* = 0.206, Language *F*(1, 46) = 0.00, *p* = 0.992, nor a significant interaction, *F*(1, 46) = 0.29, *p* = 0.591.

## 3. Experiment 2

Experiment 1 demonstrates that viewing audiovisual speech does not enhance retention of words when tested in an auditory-only modality. In Experiment 2, we explored whether the inclusion of visual speech information during retention would facilitate access to the newly learned word.

### 3.1. Participants

A new set of forty-eight 24-month-old toddlers (24 females, 24 males) took part in the study. Of these, 24 were being raised monolingual (M_age_ = 24 months 3 days, range: 1; 11; 13–2; 0; 16) and 24 were bilingual (M_age_ = 24 months 0 days, range: 1; 11; 12–2; 0; 18). See Table 1 and Table 2 for vocabulary scores and average language exposure across experiments. An additional 8 infants were tested but not included due to fussiness (2), parental interference (1), and equipment failure (5). All participants were born full term (≥37 weeks) and had no known hearing or vision problems. Families were recruited in a similar manner to those in Experiment 1.

### 3.2. Measures and Procedures

The procedure and measures were identical to those in Experiment 1, except that the toddlers saw a video of the speaker for all trial types (familiar, disambiguation, and retention trials). The videos during the retention trials consisted of the same speaker as in the familiar and disambiguation trials, producing the same tokens as the retention trials in Experiment 1 (modi and dofa). Attention to the mouth during retention trials was also considered in our exploratory analyses (recall that these trials had two novel objects on screen).

### 3.3. Results

#### 3.3.1. Familiar Trials

One-sample *t*-tests indicated that both monolingual and bilingual toddlers’ looking increased to the target object following the labeling event, *t*(23) = 3.78, *p* = 0.001, *d* = 0.772, and *t*(23) = 4.36, *p* < 0.001, *d* = 0.891, respectively (Figure 3). Independent *t*-tests comparing monolinguals and bilinguals revealed no significant difference, *t*(46) = 0.123, *p* = 0.076, *d* = 0.035.

#### 3.3.2. Disambiguation Trials

One-sample *t*-tests indicated that both monolingual and bilingual toddlers’ looking increased to the target object following the labeling event, *t*(23) = 2.15, *p* = 0.042, *d* = 0.440, and *t*(23) = 2.28, *p* = 0.036, *d* = 0.455, respectively. Independent *t*-tests comparing monolinguals and bilinguals revealed no difference in looking to the target object *t*(46) = 0.04, *p* = 0.427.

We again investigated the role of vocabulary in disambiguation. English vocabulary was significantly correlated with monolinguals’ performance in the disambiguation trials, *r*(24) = 0.44, *p* = 0.031, but not for bilinguals, *r*(24) = 0.04, *p* = 0.841. As in Experiment 1, this held true for bilinguals even when total vocabulary across both languages was considered *r*(24) = 0.10, *p* = 0.651. Translation equivalents also were not correlated with disambiguation scores for bilinguals, *r*(24) = 0.18, *p* = 0.398.

#### 3.3.3. Retention Trials

One-sample *t*-tests indicated that both monolingual and bilingual toddlers’ looking increased to the target object following the labeling event. However, this effect was only significant for bilingual toddlers, *t*(23) = 2.17, *p* = 0.040, *d* = 0.443. Monolinguals showed a marginal but non-significant difference, *t*(23) = 1.89, *p* = 0.072, *d* = 0.386. Independent *t*-tests comparing monolingual and bilinguals revealed no difference in looking to the target object, *t*(46) = 0.50, *p* = 0.517.

For exploratory analyses, we measured whether looking to the mouth during retention trials was correlated with retention scores. Both monolingual and bilingual toddlers’ retention scores were correlated with the amount of fixation to the mouth during retention trials, *r*(23) = 0.46, *p* = 0.025 and *r*(23) = 0.49, *p* = 0.015, respectively.

#### 3.3.4. Attention to Mouth

A mixed-measures ANOVA with within-factor Trial Type (mouth NL, mouth PWL, mouth retention) and between-factor Language (monolingual vs. bilingual) yielded a main effect of Trial Type, *F*(2, 90) = 7.87, *p* = 0.001, partial *n*^2^ = 0.149, and a significant Language x Trial Type interaction, *F*(1, 46) = 5.65, *p* = 0.005, partial *n*^2^ = 0.112. There was no main effect of Language, *F*(1, 45) = 0.47, *p* = 0.497.

To explore the Language x Trial Type interaction, we looked at monolinguals and bilinguals independently using paired-sample *t*-tests (with Bonferroni correction; sig. = *p* < 0.017). For monolingual toddlers, paired-sample *t*-tests revealed that there was no difference in attention to the mouth for No Learning (in which two familiar objects were on the screen) and Potential Word Learning Trials, *t*(23) = 0.72, *p* = 0.477, or No Learning and Retention trials, *t*(23) = 0.56, *p* = 0.579. There was a marginal but not significant difference between Potential Word Learning and Retention trials, *t*(23) = 1.90, *p* = 0.071, *d* = 0.393.

For bilingual toddlers, paired-sample *t*-tests revealed that there was a significant difference in attention to the mouth between No Learning and Potential Word Learning Trials, *t*(23) = 5.62, *p* < 0.001, and between No Learning and Retention trials, *t*(23) = 3.08, *p* = 0.006, with higher scores for PWL and Retention trials, respectively. No significant difference was observed between Potential Word Learning and Retention trials, *t*(23) = 1.22, *p* = 0.235.

## 4. Experiment 3

We carried out one final experiment to determine whether the effects found in Experiment 2 were driven by the visual articulatory information, or whether it was simply due to having consistent modalities across disambiguation and retention trials. In Experiment 3, toddlers saw a picture of a still face during the labeling phase of all three trial types. Thus, the same visual information was present across the trial types, but the extra-linguistic support provided by lip movements was absent. In this experiment, we opted to only test bilingual toddlers, as the results of Experiment 2 were more robust for bilinguals than monolinguals. Additionally, at the time this experiment was run, there had been no prior studies addressing fast-mapping in bilingual participants (Kalashnikova et al. (2018) was published during this time); thus, a control was deemed most urgent with bilingual participants.

### 4.1. Participants

A new set of twenty-four bilingual 24-month-olds took part in this experiment (12 females, 12 males; M_age_ = 24 months 1 day, range: 1; 11; 14–2; 0; 17). See Table 1 and Table 2 for vocabulary scores and average language exposure across experiments. An additional 3 infants were tested but not included due to fussiness (2) and equipment failure (1). All participants were born full term (≥37 weeks) and had no known hearing or vision problems. Families were recruited in a similar manner to those in Experiments 1 and 2.

### 4.2. Measures and Procedures

The materials and procedure were identical to those in Experiment 2, except that the toddlers saw a still picture of the speaker during all trial types (familiar, disambiguation, and retention trials).

### 4.3. Results

#### 4.3.1. Familiar Trials

One-sample *t*-tests indicated that toddlers’ looking increased to the target object following the labeling event, *t*(23) = 2.71, *p* = 0.012, *d* = 0.539 (Figure 4).

#### 4.3.2. Disambiguation Trials

One-sample *t*-tests indicated that toddlers’ looking increased to the target object following the labeling event, *t*(23) = 2.09, *p* = 0.048, *d* = 0.424.

As in the previous experiments, English vocabulary (*r*(24) = 0.01, *p* = 0.974), total vocabulary (*r*(24) = 0.04, *p* = 0.870), and the number of translation equivalents (*r*(24) = 0.06, *p* = 0.771) were not significantly correlated with bilinguals’ performance in the disambiguation trials for bilingual toddlers.

#### 4.3.3. Retention Trials

A one-sample *t*-test indicated bilingual toddlers’ looking did not increase to the target object following the labeling event, *t*(23) = 0.58, *p* = 0.569.

Additionally, we measured whether looking to the mouth during retention trials was correlated with retention scores as it was in Experiment 2; however, there was no significant correlation, *r*(24) = 0.12, *p* = 0.570.

#### 4.3.4. Attention to Mouth

A one-way repeated measures ANOVA was carried out, with the within-factor Trial Type (mouth NL, mouth PWL, and mouth retention) showing that there were no main effects of attention to the mouth across any trial types, *F*(2, 46) = 1.79, *p* = 0.178.

## 5. Discussion

The present work investigated the role of visual speech information in the retention of fast-mapped words. To answer this question, we manipulated the modality in which words were presented during a fast-mapping task. The results indicate that fast-mapped object–referent pairs were retained when the labels were presented in an audiovisual modality, but only if they were also tested in an audiovisual modality (as in Experiment 2). Importantly, in Experiment 2, the degree to which toddlers attended to the mouth during retention trials was directly associated with their performance in retention trials; that is, the more they attended to the speaker’s mouth, the greater likelihood that they retained the new label. Furthermore, in Experiment 3, simply seeing a face across all three trial types did not lead to retention. These results together suggest that the presence of visual information during speech processing enhances the learning and recognition of fast-mapped words. Why the visual speech information improved retention is an open question for future research. Attention to the lips during disambiguation trials was not correlated with retention scores (Experiments 1 and 2); however, attention to the lips during retention trials was positively correlated with retention scores (Experiment 2). Thus, attention to the mouth does not fully account for the improvements observed in fast-mapping. Factors such as audiovisual synchrony, improved overall attention to the stimuli, or even the mere availability of more information could also play a role in the current study.

Toddlers succeeded in the retention trials of Experiment 2 when the speaker’s lip movements were available, but not in the auditory-only retention trials of Experiment 1. This finding is reminiscent of Havy et al. (2017), in which 30-month-olds failed to recognize a visually learned word when it was only heard [35]. However, in this study, the disambiguation trials in both experiments contained both *auditory* and *visual* information, but learning did not appear to extend to the auditory-only domain (Experiment 1). We posit that task complexity likely played a role, as the current study involved an indirect word learning scenario. As discussed earlier, indirect scenarios have been shown to be a more challenging word learning condition than ostensive scenarios at 24 months [10,14].

To account for the present findings, we suggest that at 24 months, auditory information on its own—while sufficient for disambiguation—may not be sufficient to support word learning in a fast-mapping procedure (consistent with [10,14]). This account is reinforced by previous work showing that additional auditory familiarization to a novel wordform prior to a fast-mapping task did not lead to retention in 24-month-old monolinguals. On the other hand, additional exposure to the object itself prior to the experiment did facilitate retention [15]. In the current study, visual speech information supported and augmented the acoustic information, leading to a fragile word representation that is dependent on both auditory and visual information for recognition. This is consistent with other work suggesting that visual speech information may be part of the lexical representation and can therefore influence lexical access [32]. In reminder, audiovisual speech in only the disambiguation phase was not sufficient for retention (Experiment 1). Furthermore, a still face during disambiguation and testing (Experiment 3) did not lead to retention. It was only in Experiment 2, when dynamic auditory–visual information was presented during both the initial disambiguation trials and retention trials, that retention—particularly for the bilingual infants—was boosted. Moreover, in Experiment 2, looking to the mouth during retention trials was positively correlated with increased looking to the target object. Again, the mechanisms driving this behavior could be any combination of attention, audiovisual synchrony, and/or more available linguistic information. Regardless, these results demonstrate that the presence of visual speech information was necessary both during disambiguation and retention trials for successful performance in the retention trials. These findings together reinforce the conclusion that visual speech information plays a powerful role in the learning and recognition of fast-mapped words.

This explanation is also informative for our understanding of the developmental trajectory of bilingual children’s ability to form and retain novel word–referent mappings. Previous work has suggested that bilinguals are less likely to employ disambiguation than their monolingual peers [18,19,54,55], or are more selective in the instances in which fast-mapping is employed, in order to allow for lexical overlap (e.g., [24]). A dynamic associative account of word learning would suggest that bilingual toddlers’ vocabulary in *two* languages makes fast-mapping tasks more difficult than for their monolingual peers. As such, the mechanisms that underlie bilinguals’ slow associative learning, which ultimately lead to retention in a fast-mapping task, may require additional linguistic support [13]. In the current study, this additional linguistic support is in the form of visual speech information. Indeed, we see improvements in bilinguals’ retention when visual speech information is present during disambiguation and retention trials (and monolinguals for that matter). Thus, additional linguistic support improved performance in a fast-mapping task, consistent with a dynamic associative account. This does not necessarily rule out a lexical constraints account for fast-mapping, as a lexical constraints account could argue that disambiguation as a word learning heuristic will operate if sufficient information is available, but it does leave an open question for future research.

One limitation of this work is that bilingual children’s vocabulary was assessed using an English CDI form that the parents translated themselves. While this was done for practical reasons (as there are not CDIs available for every language, and we used a sample of bilinguals learning a variety of different languages), this method may underestimate bilinguals’ knowledge in their other language. That is, an early learned word in English may not be an early learned word in another language. This may have had an impact on both base CDI numbers as well as the estimation of translational equivalents. This may have had an influence on why there was no correlation between bilinguals’ vocabulary, or translational equivalents, with performance on disambiguation trials. An additional limitation of this work is whether the proximity of bilinguals’ languages has an effect on children’s attention to the speaker’s mouth. While the effect of proximity of languages might be task-dependent, the present study draws from a diverse group of bilinguals that might obscure any language proximity effects (see Appendix A). Some studies with visual language have shown similar results at 8 months with infants learning either close or distant languages, even when presented with the same stimuli [2,56]. On the other hand, recent results on audiovisual speech perception find language proximity effects in looking behavior to the mouth [33]. Thus, the effect of bilinguals’ language proximity in looking behavior during word retention is one avenue worth exploring for future work.

## 6. Conclusions

Overall, the present work has implications for the development of the monolingual and bilingual lexicon. We demonstrate that visual speech information influenced the retention of fast-mapped words in 24-month-old monolingual and bilingual toddlers. This work strengthens the claim that fast-mapping is an available heuristic to infants this young and shows that visual information in talking faces does augment an auditory-only signal, providing additional linguistic information in support of the learning and recognition of novel words.

## Figures and Tables

**Figure 1 brainsci-11-00114-f001:**
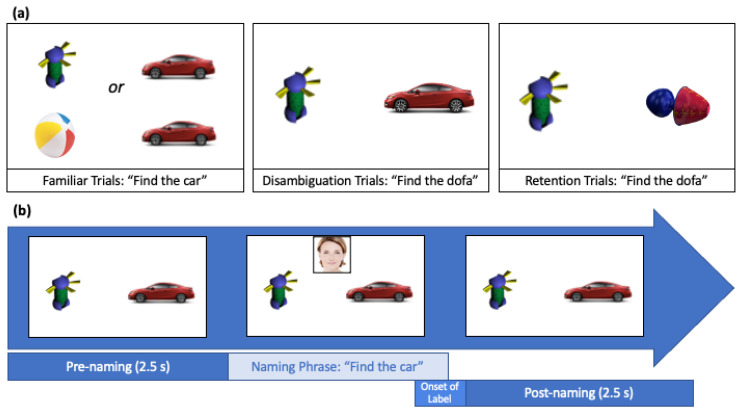
(**a**) Top row displays examples of the familiar, disambiguation, and retention trials. Note that these are not the stimuli as they appeared in the study. (**b**) Bottom row displays the time course of each trial, in which there was a pre-naming period (2.5 s), followed by the naming phrase, followed by the post-naming period (2.5 s).

**Figure 2 brainsci-11-00114-f002:**
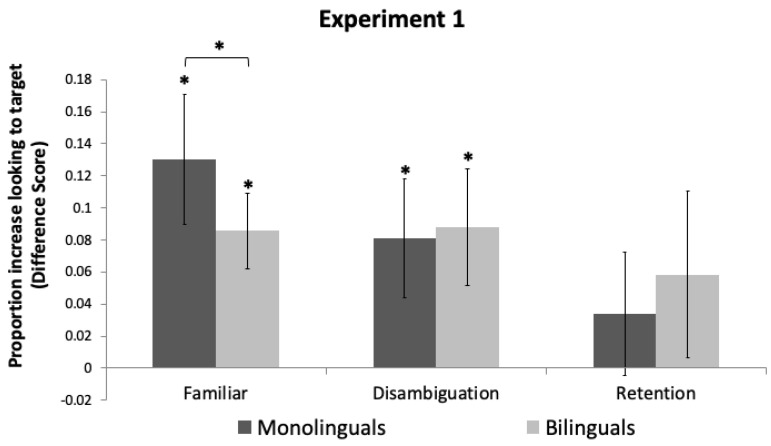
Difference scores and standard errors for Experiment 1. Positive scores indicate increased looking to the target object. * indicates a *p*-value smaller than 0.05.

**Figure 3 brainsci-11-00114-f003:**
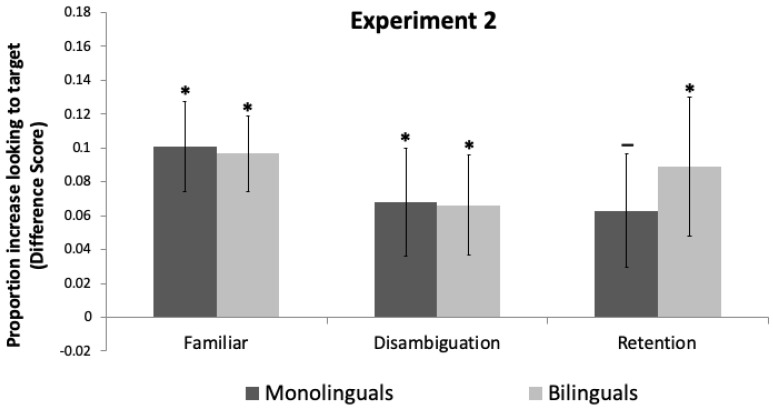
Difference scores and standard errors for Experiment 2. Trial type is on the *X*-axis. The *Y*-axis gives the proportion of looking to the target object post-naming–pre-naming periods. Positive scores indicate increased looking to the target object. * indicates a *p*-value smaller than 0.05, —indicates a marginal but non-significant *p*-value.

**Figure 4 brainsci-11-00114-f004:**
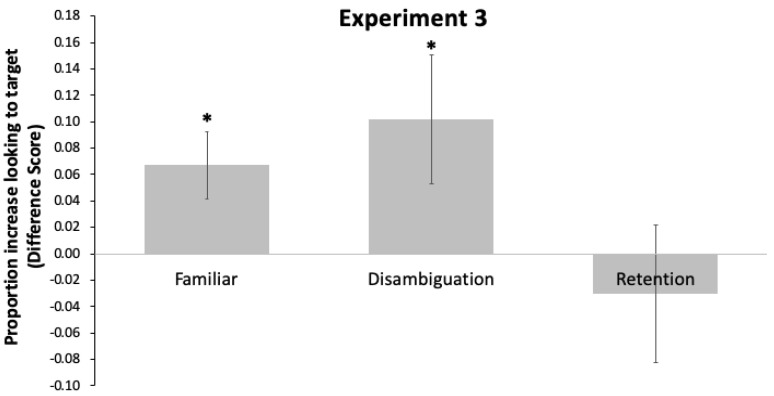
Difference scores and standard errors for Experiment 2. Trial type is on the *X*-axis. The *Y*-axis gives the proportion of looking to the target object post-naming–pre-naming periods. Positive scores indicate increased looking to the target object. * indicates a *p*-value smaller than 0.05.

**Table 1 brainsci-11-00114-t001:** Maternal and paternal education levels for participants for all three experiments. Parents indicated which of the following options best described their highest level of education: (1) below high school, (2) high school, (3) some college or certification, (4) Bachelor’s degree, (5) Master’s degree, (6) professional degree (e.g., PhD, MD). We report the average, median, and range.

	*Average*		*Median*		*Range*		*Data Not Available*
	Maternal	Paternal	Maternal	Paternal	Maternal	Paternal	
*Experiment 1*	4.2	4.1	4	4	3–6	2–6	4 participants
*Monolingual*	4.2	4.2	4	4	3–6	2–6	
*Bilingual*	4.2	4.0	4	4	3–6	2–6	
*Experiment 2*	4.6	4.2	4	4	2–6	1–6	3 participants
*Monolingual*	4.8	4.7	5	5	3–6	2–6	
*Bilingual*	4.3	3.8	4	4	2–6	1–6	
*Experiment 3*	4.5	4.5	4	4.5	2–6	2–6	NA

**Table 2 brainsci-11-00114-t002:** Descriptive statistics for the CDI (Communicative Development Inventory) information for all experiments. The values represent the Average Total English Vocabulary (standard deviation), Average Translation Equivalents ^1^, and Average Total Estimated Vocabulary in Both Languages (English Vocabulary + Translation Equivalents).

		English Vocabulary	Translation Equivalents	Total Estimated Vocabulary
Experiment 1	Monolingual	41 (20)	1 (2)	42
	Bilingual	38 (16)	6 (8)	44
Experiment 2	Monolingual	49 (29)	1 (3)	50
	Bilingual	54 (16)	4 (4)	58
Experiment 3	Monolingual	-	-	-
	Bilingual	42 (21)	12 (13)	54

^1^ A few monolingual children had learned one to three words in another language through occasional, but not systematic, exposure to that language (e.g., learned from a book or a family friend).

**Table 3 brainsci-11-00114-t003:** Language exposure information for all experiments. Values represent the overall percent exposure to English (standard deviation). The number of unique languages that bilinguals were exposed to is also provided (note that each child heard only one of these languages; see Appendix A).

Average Exposure to English (%)	Experiment 1	Experiment 2	Experiment 3
Monolingual	96.6% (4.5)	96.8% (3.9)	
Bilingual	58.5% (10.1) 18 languages	54.5% (16.4) 15 languages	54.8% (14.5) 15 languages

## Data Availability

The data presented in this study are available online at https://osf.io/c8kn9/?view_only=7b6f5f0c436745ae90a50d1017a1e276.

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
