# Peer review of "The Role of Audiovisual Speech in Fast-Mapping and Novel Word Retention in Monolingual and Bilingual 24-Month-Olds"

_brainsci, 2021, doi:10.3390/brainsci11010114_

Round 1

Reviewer 1 Report

This article presents the results of 3 experiments that investigated the role of visual speech cues on monolingual and bilingual toddlers’ use of the disambiguation heuristic for selecting the referents of novel words and retention of these novel mappings. Results showed that monolingual and bilingual toddlers relied on disambiguation and showed retention when novel words were presented by a speaking face. However, infants were not successful at retention when the speaking face appeared only in the disambiguation trials of the task or when a static face appeared in all trials. Infants’ reliance on visual information for retention was further supported by the finding of a significant correlation between infants’ tendency to look at the speaker’s mouth and their retention performance.   This work investigates a long-standing question regarding the role of the disambiguation heuristic in early word learning, and importantly, the effects of infants’ language experience on the emergence and use of early word-learning strategies. Its contribution to the existing literature is novel and timely - there continues to be scarce evidence on the role of visual speech cues on early speech processing and word learning. The experiments are carefully designed and include a total of 120 toddlers, which is a good indication of the robustness of its findings. Overall, the articule is succinct and clear and provides clear motivation for the experiments. I have several comments that can be addressed in a revision, mainly focusing on details about the method and analyses.   
  1. Did the bilingual infants from the three experiments receive any exposure to a third language? If so, please provide this information in the text or Table 2. 
  2. The Method section mentions that information about maternal education was collected, but this is not reported. It is important for transparency to see an index of SES and evidence that it did not differ between the monolingual and bilingual groups. 
  3. In Table 1, what is ’total estimated vocabulary’? This measure does not appear to be mentioned in the Method or the Results. 
  4. What was bilinguals’ vocabulary size across the two languages? Since this measure was included in the correlations, please provide descriptive stats in Table 1. 
  5. There were 2 novel objects, but 6 familiar objects in the task. Were the same two familiar objects used in all disambiguation trials, or were all 6 used across trials?
  6. One important omission in the paper is that there is no description of how the eye-tracking data were processed. What were the rates of gaze loss? Was any software used to process the eye-tracking data?
  7. Calculation of proportion of looks to the mouth: the text states that this proportion was calculated out of the fixation time to the mouth and the two objects. What about fixations to other parts of the face? Did the children fixate the eyes of the speaker? If so, why was this not accounted for in any of the analyses? 
  8. I found the description of the PWL trials very confusing. The description of how proportion of looks to the mouth were calculated states that this was done to compare Familiar and Disambiguation trials, so when PWL trials are introduced, it sounds like this would be an additional analysis. In the Results, however, it becomes clear that only Familiar and PWL trials were compared. Then, the number of these trials is also not clear. The task included 12 disambiguation trials (I understand that these were all PWL), 4 familiar trials that contained one familiar and one novel object (so also PWL), and 4 familiar trials that contained two familiar objects (so familiar trials for this purpose). This would mean that these analyses compared performance on 16 PWL and 4 familiar trials. Is this correct? It would be helpful to see these details in the paper. It may be clearer to mention the proportions to the mouth in this section and specify that gaze to the mouth on 16 familiar - novel trials (regardless of the label) was compared to 4 familiar-familiar trials. 
  9. Throughout the Results section, p-values over .07 are interpreted as trends. It is contentious to do so given that the p-value is not the best indication of whether there were or not meaningful differences between the groups. For instance, the effect size for the ’trend’ for the familiar trials in Expt. 2 is .035, which suggests that it is unlikely that there was a difference. I would suggest that the authors base their interpretations on effect sizes for interpreting these potential trends.
  10. Related to the above, please provide the effect size for the t-test on p. 10, line 384.
  11. Please specify whether p-values were adjusted for multiple comparisons for the pairwise t-tests conducted to investigate the interaction in Expt. 2 (attention to the mouth). If not, please provide a justification. 
  12. Throughout the manuscript, there is no justification for why only bilinguals were included in Experiment 3. I am not suggesting that a monolingual group is needed, but it is important to see this explained in the paper. 

Author Response

We thank the reviewer for their thoughtful review. We address each of their concerns below.

  1. Did the bilingual infants from the three experiments receive any exposure to a third language? If so, please provide this information in the text or Table 2. 

Children in this study received no systematic exposure to a third language. We have added this to the manuscript line 228.

  1. The Method section mentions that information about maternal education was collected, but this is not reported. It is important for transparency to see an index of SES and evidence that it did not differ between the monolingual and bilingual groups. 

We have added the requested information to our demographics section of Experiment 1, starting on line 215:

Parents reported on their child’s health, birth weight, sex, and gestational age. Parents also reported their level of education, work status, and racial or ethnic background. In all three experiments, parental education levels were on average between a Bachelor’s degree and a Master’s degree. See Table 1 for a full summary.

Table 1. Maternal and Paternal Education levels for participants for all three experiments. Parents indicated which of the following options best described their highest level of education: (1) below high school, (2) high school, (3) some college or certification, (4) Bachelor’s degree, (5) Master’s degree, (6) professional degree (e.g., PhD, MD). We report the average, median, and range.

Average

Median

Range

Data not available

Maternal

Paternal

Maternal

Paternal

Maternal

Paternal

EXPERIMENT 1

Monolingual

Bilingual

4.2

4.2

4.2

4.1

4.2

4.0

4

4

4

4

4

4

3-6

3-6

3-6

2-6

2-6

2-6

4 participants

EXPERIMENT 2

Monolingual

Bilingual

4.6

4.8

4.3

4.2

4.7

3.8

4

5

4

4

5

4

2-6

3-6

2-6

1-6

2-6

1-6

3 participants

EXPERIMENT 3

4.5

4.5

4

4.5

2-6

2-6

NA

  1. In Table 1, what is ’total estimated vocabulary’? This measure does not appear to be mentioned in the Method or the Results. 

It is the total number of words in both their two languages. English + Translation Equivalents. We have added this description to the manuscript line 231,

  1. What was bilinguals’ vocabulary size across the two languages? Since this measure was included in the correlations, please provide descriptive stats in Table 1. 

See above comment.

  1. There were 2 novel objects, but 6 familiar objects in the task. Were the same two familiar objects used in all disambiguation trials, or were all 6 used across trials?

All six were used across trials. We have clarified this point in the manuscript see line 282.

  1. One important omission in the paper is that there is no description of how the eye-tracking data were processed. What were the rates of gaze loss? Was any software used to process the eye-tracking data?

We have now included this information in the Gaze Coding section of Experiment 1, starting line 293:

All eye-tracking data was processed using the SMI BeGaze software. The raw average rate of gaze loss during trials was 14.3% across all experiments (14.6%, 14.1%, and 14.3% for Experiments 1, 2, and 3 respectively). Trials in which there was no eye-tracking data during either the pre-naming or post-naming phase were excluded from our analysis.

  1. Calculation of proportion of looks to the mouth: the text states that this proportion was calculated out of the fixation time to the mouth and the two objects. What about fixations to other parts of the face? Did the children fixate the eyes of the speaker? If so, why was this not accounted for in any of the analyses? 

We would like to clarify that our main hypothesis was that the presence of the face would improve performance in the fast-mapping task. We did analyse the mouth as a region of interest for exploratory analyses, but we had no a priori expectations about children’s looking to the eyes. Our region of interest was the mouth, and the two objects on the screen. The eyes were not considered a region of interest and this data was not recorded or analyzed.

  1. I found the description of the PWL trials very confusing. The description of how proportion of looks to the mouth were calculated states that this was done to compare Familiar and Disambiguation trials, so when PWL trials are introduced, it sounds like this would be an additional analysis. In the Results, however, it becomes clear that only Familiar and PWL trials were compared. Then, the number of these trials is also not clear. The task included 12 disambiguation trials (I understand that these were all PWL), 4 familiar trials that contained one familiar and one novel object (so also PWL), and 4 familiar trials that contained two familiar objects (so familiar trials for this purpose). This would mean that these analyses compared performance on 16 PWL and 4 familiar trials. Is this correct? It would be helpful to see these details in the paper. It may be clearer to mention the proportions to the mouth in this section and specify that gaze to the mouth on 16 familiar - novel trials (regardless of the label) was compared to 4 familiar-familiar trials. 

We apologize, we realize using the term Familiar lead to confusion in this description. We have clarified our intentions and analyses in a few places in the manuscript.

First, we have better set up the rationale for these analysis in the Current Study portion of the introduction, starting at line 173:

Finally, as previous work has suggested that infants and toddler’s attention to the mouth is driven by an attempt to learn language [35-40], we recorded children’s looking to the speaker’s mouth when present for the purpose of exploratory analyses. The rational being that greater attention to the mouth will be paid when an unfamiliar object(s) is on screen, as there is an opportunity for learning (Disambiguation trials, Experiment 1 and 2), and additional information from the mouth may be needed for accurate recognition (Retention Trials, Experiment 2). We assessed whether there were correlations between looking to the mouth when there were 0, 1, and 2 unfamiliar objects on the screen and children’s retention of the fast-mapped words.

In the description of the Exploratory Analysis, we have changed the wording to clearer, and also specified the number of trials as suggested, starting line 315:

For the purposes of investigating exploratory correlations between looking to the mouth and looking to the target object, we distinguish between two types of trials in which attention was paid to the mouth: trials in which there were no novel objects on the screen (i.e., familiar trials in which there were two known objects such as book and cup) called No Learning trials (NL), and trials where there was one novel object on the screen (Familiar and Disambiguation trials in which there was one familiar object and one novel object on the screen) called Potential Word Learning trials (PWL). In total there were 4 No Learning Trials and 16 Potential Word Learning Trials. Attention to the mouth during No Learning and Potential Word Learning Trials was compared to determine if there were any overall differences across trial types. Furthermore, correlations between looking to the mouth during PWL trials and performance on Disambiguation and Retention trials were run.

In the Attention to the Mouth analyses section for Experiments 1, 2, and 3, we changed all instances of Familiar to No Learning (see lines 356, 418, and 475 for the start of each section).

  1. Throughout the Results section, p-values over .07 are interpreted as trends. It is contentious to do so given that the p-value is not the best indication of whether there were or not meaningful differences between the groups. For instance, the effect size for the ’trend’ for the familiar trials in Expt. 2 is .035, which suggests that it is unlikely that there was a difference. I would suggest that the authors base their interpretations on effect sizes for interpreting these potential trends.

We have removed the word trend altogether. In the case of marginal non-significant differences that have small effect sizes (i.e., cohens d >.2) we used the wording “a marginal but non-significant difference” and reported the cohens d (see lines 411 and 427). In the case of the example the reviewer provided (line 396) we used the wording “Independent t-tests comparing monolingual and bilinguals revealed no significant difference”.

  1. Related to the above, please provide the effect size for the t-test on p. 10, line 384.

We have provided this effect size.

  1. Please specify whether p-values were adjusted for multiple comparisons for the pairwise t-tests conducted to investigate the interaction in Expt. 2 (attention to the mouth). If not, please provide a justification. 

We apologize for not stating this in the paper. A Bonferroni correction was used (sig. = p < .17). We have added this information to the manuscript on line 424.

  1. Throughout the manuscript, there is no justification for why only bilinguals were included in Experiment 3. I am not suggesting that a monolingual group is needed, but it is important to see this explained in the paper. 

Reviewer 2 Report

This is a very interesting and well written paper about how visual speech cues (lip movements produced during child directed speech) support early learning and retention of words in monolingual and bilingual toddlers. The manuscript was a pleasure to read and there are only a few relatively minor suggestions.

The transition to the key topic of visual linguistic support would benefit from more development. The focus on this particular cue emerges rather suddenly. It seems, for example, that visual speech cues may provide important scaffolding when there are two phonological forms for the same meaning, especially when these two forms are phonologically similar. This point is just illustrative that some more explanation for why this particular support was selected for study would be helpful.

Pg. 3, typo, “based solely based on…..”

As presented in the introduction, it seems that the relationship between the two languages being learned is important—e.g., languages with more similarities in phonological structure or in word form-referent pairings may be treated differently from those with fewer similarities (e.g., Spanish vs. Mandarin). In the actual studies, a wide range of languages are included. This point would merit some attention.

Was there any information about age of exposure to both languages? Were infants generally exposed to both languages from birth?

Are visual features of the novel objects (or, for that matter, the familiar objects) controlled in any way. For size? Color? Similarity? Potential function?

Is looking to the mouth expected to be contingent on whether the word is novel? Do infants look more when a word is unfamiliar? Is that the rationale for the comparison? This becomes clear in the experiments, but a little more set up in the introduction would be helpful.

A little bit more methodological detail in experiment 2 would benefit the reader-- regarding the visual input during the retention phase. I assume it’s the same video of the same talker producing the same tokens.

Overall, this paper was a delight to read and it tells a very interesting story.

Author Response

We thank the reviewer for their kind words and helpful suggestions. We have detailed our responses to their points below:

The transition to the key topic of visual linguistic support would benefit from more development. The focus on this particular cue emerges rather suddenly. It seems, for example, that visual speech cues may provide important scaffolding when there are two phonological forms for the same meaning, especially when these two forms are phonologically similar. This point is just illustrative that some more explanation for why this particular support was selected for study would be helpful.

We thank the Reviewer for this suggestion. To address this concern, we have added a line to transition between the topic of fast mapping into audiovisual cues (see lines 91-92), as well as more information on how visual cues scaffold speech processing when similar phonological forms are encountered (see lines 99-105).

Pg. 3, typo, “based solely based on…..”

This has been corrected.

As presented in the introduction, it seems that the relationship between the two languages being learned is important—e.g., languages with more similarities in phonological structure or in word form-referent pairings may be treated differently from those with fewer similarities (e.g., Spanish vs. Mandarin). In the actual studies, a wide range of languages are included. This point would merit some attention.

Was there any information about age of exposure to both languages? Were infants generally exposed to both languages from birth?

In almost all cases children were exposed to both languages right from birth; however, the proportion that they heard each language may change over time. Unfortunately, this information was not consistently recorded across experiments and thus we are unable to report it.

Are visual features of the novel objects (or, for that matter, the familiar objects) controlled in any way. For size? Color? Similarity? Potential function?

We have included more information regarding our visual stimuli starting on line 262:

The visual stimuli were pictures of six familiar objects that represented some of the first words learned by infants exposed to English (car, cup, book, ball, cookie, shoe) and two novel objects (see Bion et al., 2011). The two novel objects were images downloaded from the TarrLab Object DataBank [49], with the criteria that they did not resemble any familiar objects in our stimuli, or any other likely known objects at this age. All objects were controlled for size and perceptual similarity. Arrays of two objects were presented on a grey background, these objects were automatically located by the SMI begaze software in the middle-left or middle-right positions of the screen, at the same distance from the centre of the screen (See Figure 1).  

Is looking to the mouth expected to be contingent on whether the word is novel? Do infants look more when a word is unfamiliar? Is that the rationale for the comparison? This becomes clear in the experiments, but a little more set up in the introduction would be helpful.

We have better set up the rationale for these analysis in the Current Study portion of the introduction, starting at line 173:

Finally, as previous work has suggested that infants and toddler’s attention to the mouth is driven by an attempt to learn language [35-40], we recorded children’s looking to the speaker’s mouth when present for the purpose of exploratory analyses. The rational being that greater attention to the mouth will be paid when an unfamiliar object(s) is on screen, as there is an opportunity for learning (Disambiguation trials, Experiment 1 and 2), and additional information from the mouth may be needed for accurate recognition (Retention Trials, Experiment 2). We assessed whether there were correlations between looking to the mouth when there were 0, 1, and 2 unfamiliar objects on the screen and children’s retention of the fast-mapped words.

A little bit more methodological detail in experiment 2 would benefit the reader-- regarding the visual input during the retention phase. I assume it’s the same video of the same talker producing the same tokens.

We have added more information regarding the videos starting line 382:

The videos during the Retention trials consisted of the same speaker as in the Familiar and Disambiguation trials, producing the same tokens as the Retention trials Experiment 1 (modi and dofa).